# Influence of Ripening on Chemical Characteristics of a Traditional Italian Cheese: Provolone del Monaco

**Nadia Manzo [1], Antonello Santini [2,\*]** **, Fabiana Pizzolongo [1]** , **Alessandra Aiello [1]** ,
**Andrea Marrazzo [1], Giuseppe Meca [3], Alessandra Durazzo [4], Massimo Lucarini [4] and**
**Raffaele Romano [1]**

[1] Department of Agriculture, University of Napoli Federico II, Via Università, 100-80055 Portici (Napoli), Italy; nadia.manzo@unina.it (N.M.); fabpizzo@unina.it (F.P.); alessandra.aiello@unina.it (A.A.); draven88@hotmail.it (A.M.); rafroman@unina.it (R.R.)
[2] Department of Pharmacy, University of Napoli Federico II, Via D. Montesano, 49-80131 Napoli, Italy
[3] Laboratory of Food Chemistry and Toxicology, Faculty of Pharmacy, University of Valencia, Av. Vicent Andrés Estellés s/n, 46100 Burjassot, Spain; giuseppe.meca@uv.es
[4] CREA—Research Centre for Food and Nutrition, Via Ardeatina 546, 00178 Roma, Italy; alessandra.durazzo@crea.gov.it (A.D.); massimo.lucarini@crea.gov.it (M.L.)
\* Correspondence: asantini@unina.it; Tel.: +39-81-253-9317

**Abstract:** The envisaged promotion of local products contributes to environmental protection and is a valid tool for the promotion of socioeconomic development, enhancement of territories, and biodiversity preservation and sustainability. Provolone del Monaco is a semi hard pasta filata cheese granted PDO (Protected Designation of Origin) designation by the European Union. Provolone del Monaco is obtained from raw cow's milk, produced in the specific areas of the Lattari Mountains and Sorrento Peninsula (Naples, Italy), and ripened for at least six months. To the best of our knowledge, no studies concerning the complete chemical characterization of Provolone del Monaco cheese are available. In the present study; the chemical characterization (moisture; pH; titratable acidity; nitrogen; and fat content), fatty acid composition determined by using gas-chromatography-flame-ionization-detector (GC-FID); volatile organic compounds by solid-phase microextraction followed by gas chromatography-mass spectrometry (SPME-GC/MS), and maturation indices were evaluated during ripening. Two different average typical cheese sizes (3 kg and 5 kg) and two different internal portions were studied. After 6 months of ripening, the most important changes recorded were the loss of water, the increase in acidity, the nitrogen (as ammonia) release, and the production of volatile organic compounds. The cheese size did not affect the chemical composition of Provolone del Monaco.

**Keywords:** Provolone del Monaco; traditional foods; biodiversity; sustainability

## 1. Introduction

The great progress in technological processes, agricultural practices, and changes in life style has led toward paying greater attention to local products as principal elements for food product quality improvement, and in the meantime, supporting local agro-biodiversity. The role of biodiversity for food quality and healthy status is well recognized: traditional, local, and season foods are produced by techniques based on historic and cultural traditions of a specific territory and which occur only in a local place. The promotion of local foods is well addressed towards a sustainable and environmentally friendly production system, as well as to validated traceability systems. An appropriate use of farmland, protection of animal health and welfare, and environmental conservation are linked to

climate knowledge, soil quality, and landscaping, leading to substantial improvements in the product quality. The envisaged promotion of local products contributes to environmental protection and it is a valid tool for the promotion of socioeconomic development, enhancement of territories, and biodiversity preservation and sustainability. There is the need to assess the extent to which it will become clear that local/traditional foods play an important role in the food pattern of many population groups in European countries. Components of traditional diets can be reinforced and promoted for their nutritional quality, health properties, and safety. From this perspective, nutritional science should support sustainable ecosystems, ecological resources, and healthy environments, because nutrition and environmental sustainability are strictly linked through the food system.

Provolone del Monaco is a semi-hard pasta filata cheese produced from raw cow's milk in the Lattari Mountains and in Sorrento Peninsula areas within the Campania Region (Italy). This cheese, shown in Figure 1, is ripened for at least six months before being considered a PDO (Protected Designation of Origin) cheese, according to the Italian Regulation set in 2010 [1].

The PDO designation indicates the particular characteristics of quality of this cheese depending on local factors, such as: (i) milk origin (the milk used for cheese making is produced by cattle living exclusively in the manufacturing area), (ii) traditional cheese making process, and (iii) microclimate conditions of the production area. This cheese is cylindrical shape; it is characterized by having a creamy white pasta filata, and a smooth and yellowish rind. The smell of Provolone del Monaco is strong and penetrant, while the taste can be sweet or spicy, depending on the ripening time.

According to the PDO regulation for the production of this cheese, an amount ≥ 20% of the milk used for making the Provolone del Monaco cheese must be obtained from cows belonging to the "Agerolese" breed, which originates from the Sorrento Peninsula area (located in the Campania Region, South of Italy). This breed produces milk with high butterfat content. The remaining 80% of the milk used for making the Provolone del Monaco cheese comes from different local cattle breeds (e.g., Frisona, Jersey, Brunalpina, and Podolica). According to the regulation for producing the Provolone del Monaco cheese, at least 40% of the dry matter feed for cows must come from fresh fodder.

Only a few studies regarding Provolone del Monaco PDO cheese are available in the scientific literature. Aponte et al. [2] focused on lactic acid bacteria occurring during cheese manufacture and ripening. Romano et al. [3] studied how ripening can influence the amounts of cheese fatty acids, $\omega-3$ and conjugated linoleic acids (CLA). Also, diet can influence the composition of milk fat [4].

The results of these studies indicated that for Provolone del Monaco, the main fatty acids present were medium molecular weight fatty acids (43%), ranging from C11:0 to C16:1 cis-9, during the first six months of ripening, and after this period of time, high molecular weight fatty acids (41%) and low molecular weight fatty acids (8%) were present. The minor acidic component, represented by the c9t11 conjugated linoleic acid (CLA), did not show significant differences during ripening. Di Monaco et al. [5] observed that Provolone del Monaco cheese sensorial attributes recognized as typical by consumers are vinegar odor, ripened flavor, and appearance of peculiarities, such as the number of holes (eyelets) of different forms and dimensions inside the cheese.

The total fat, pH, protein, and moisture contents of Provolone del Monaco cheese have a significant effect on the degree of release of volatile compounds, including several free fatty acids (FFAs), and can affect lipase activity [6,7]. Short chain fatty acids directly contribute to flavor, but fatty acids can also act as precursors for the production of a wide range of other volatile flavor compounds [8].

The principal pathways for the formation of flavor compounds in cheese during ripening are glycolysis, lipolysis, and proteolysis [8–10]. Glycolysis refers to the metabolism of residual lactose, lactate, and citrate. Lactate contributes to the flavor of acid-curd cheeses and probably also contributes to the flavor of ripened cheese varieties, particularly early in maturation. Acidification of the cheese has a major indirect effect on flavor since it determines the buffering capacity of the cheese and the growth of various microorganisms during ripening other than the activity of the enzymes involved in cheese ripening. Lactate can be oxidized to acetate and $CO_2$ by the non-starter lactic acid bacteria present in cheeses, and the availability of $O_2$ can be influenced by the size of the cheese and by the

oxygen permeability of the rind or packaging material [11]. The lipid fraction of the cheese can undergo oxidative or enzymatic hydrolytic degradation with the production of FFAs (lipolysis). The agents responsible for lipolysis in cheese made from raw milk, such as Provolone del Monaco, are indigenous milk lipase, raw milk microflora, and non-starter lactic acid bacteria. The impact of FFA on the flavor can also be influenced by the pH since carboxylic acids and their salts are perceived differently by the consumers, and a higher pH value lowers the FFA perception. Free fatty acids also act as precursor molecules for a cascade of catabolic reactions that lead to the production of other flavor compounds, such as methyl ketones, lactones, esters, alkanes, and secondary alcohols [11].

Proteolysis is the most complex biochemical reaction that occurs during cheese ripening and consists of catabolic reactions and degradation of the casein matrix to a range of peptides and free amino acids (FAAa), and of subsequent reactions involved in the catabolism of FAA. Proteolysis plays a vital role in the development of textural changes and flavor. The contribution to flavor is due to the formation of peptides and free amino acids, the liberation of substrates (amino acids) for secondary catabolic changes, and the change of the cheese matrix, which facilitate the release of sapid compounds during mastication [11].

There is a lack of studies regarding the complete chemical characterization of Provolone del Monaco cheese correlating the chemical characterization, fatty acid composition, volatile organic compounds (VOCs), and maturation index during ripening. The aim of the present paper is to fully characterize the nutritional properties of Provolone del Monaco, an example of traditional Italian cheese. Moreover, the effect of ripening, which also includes comparisons between two different cheese sizes and between two different internal parts of the cheese are evaluated.

## 2. Materials and Methods

### 2.1. Cheese-Making

The typical protocol for Provolone del Monaco production is quite long and starts by mixing together the milk collected in the morning and in the evening, after a preliminary cleaning step by centrifugation. The following steps are: (i) the coagulation (40–60 min) by adding lamb rennet paste without starter addition; (ii) curd cutting to hazelnut size first and to corn bean later; (iii) cooking at T in the range from 48 to 52 °C for 30 min; (iv) whey draining on flaxen cloths; (v) curd acidification on a wood table at room temperature for t = 12–14 h; (vi) stretching in water at T in the range 85–95 °C; (vii) molding in pieces; (viii) salting in brine (t = 8–12 h/kg); (ix) ripening at T in the range 8–15 °C, and 85% relative humidity for a minimum of 6 months. After ripening, the cheese pieces must have a weight in the range from 2.5 to 8 kg, and the ratio between fat and dry matter must be >40.5%.

### 2.2. Sampling

Three batches of Provolone del Monaco were produced in June 2016 in the factory Perrusio S.r.L. located in Meta di Sorrento (Napoli, Italy) according to the PDO regulation. For each batch, two cheese sizes, namely, of 3 and 5 kg of weight, were studied.

Three samples of both size cheeses for each batch were collected every 90 days for analysis (time 0, 90, 180, and 270 days) in a ripening chamber with controlled temperature and humidity, equipped with a data logger (Testo mod. 174H) for temperature and humidity data acquisition. The collection was repeated two times. Sample at time 0 was considered immediately after salting the cheese in brine.

In each sample, two different areas of the cheese were collected: the core (C), at 4–6 cm from the rind, and the portion just below the rind (S), at 0.3–0.5 cm from the rind. The samples were vacuum-sealed and kept in a freezer at −20 °C until the analysis.

### 2.3. pH and Moisture Content

The pH of the cheeses was determined by direct insertion of a suitable pH meter (sensION+, Hach Lange, CO, USA) in three different parts of the cheese. Moisture content was determined by

drying 5 g of sample at 102.0 ± 2.0 °C until constant weight [12]. Results were expressed as weight percentage (%, w/w).

## 2.4. Titratable Acidity

Ten grams of grated cheese was added with 50 mL of deionized water and put in a bath at 40 °C for 5 min. The mixture was homogenized for 5 min with an Ultra-turrax T25 homogenizer (Janke & Kunkel, GmbH & CO, Staufen, Germany). The suspension was transferred in a 100 mL volumetric task and made up to the mark using deionized water. The suspension was filtrated, and 25 mL was titrated using 0.1 NaOH and phenolphthalein as indicator (D.M. 1986, FIL-IDF 5A:1969). The results were expressed in percent by weight of lactic acid (% w/w).

## 2.5. Nitrogen Content

Cheese samples were analyzed using the official method Kjeldahl method [13] to determine nitrogen fractions: the total nitrogen (TN) and the pH 4.6-soluble nitrogen (SN). To determine TN, 0.6 g of grated cheese was inserted into a Kjeldahl tube of 20 mL sulfuric acid 98% and 2 g of catalyst (cupric sulfate and potassium sulfate in the ratio 1:24) was added. In order to determine SN, 20 g of grated cheese was put into a 300 mL beaker, and 80 mL of 0.5 M trisodium citrate at pH 7 was added and then kept at 40 °C for 1 h. After this time, 3 drops of formaldehyde were added and the mixture was homogenized at 24.000 rpm for 1 min. After this step, the pH was adjusted to 4.6 using HCl 37% and the mixture was brought to a volume of 500 mL using deionized water. After 1 h resting, the solution was filtrated through Whatman n°42 filter paper. Sulfuric acid and catalyst were added to 25 mL of the filtrated solution. The results were expressed in percent on dry weight (%, w/dw).

## 2.6. Fat Content

The fat content was gravimetrically determined according to the method described in D.M. 1986 [13], which is based on the Schimith–Bondzynski–Ratzlaff traditional method of extracting lipids, with some modifications. Ten grams of grated cheese were hydrolyzed using 10 mL of 37% hydrochloric acid (d = 1.125) and 7 mL of 95% (v/v) ethyl alcohol. The cheese suspension was homogenized with an Ultra-turrax T25 homogenizer (Janke & Kunkel, GmbH & CO, Staufen, Germany) in a glass beaker for 30 min at 50 °C with constant magnetic stirring. After cooling, the fatty matter was extracted using 20 mL of ethyl ether-petroleum ether 1:1 solution under constant stirring for 15 min. The suspension was then rested for 10 min to allow phase separation. This extraction protocol was repeated 3 times. Three organic extracts were pooled, dried over anhydrous sodium sulfate, filtered with a cellulose filter, evaporated under reduced pressure in a rotary evaporator and weighed. The results were expressed in percent on dry weight (%, w/dw). The fatty extracts were analyzed for fatty acids.

## 2.7. Gas Chromatographic Analysis of Fatty Acids

Fatty acid methyl esters (FAMEs) were prepared using 2 N potassium hydroxide in a methanol solution as described by Nota et al. [14], with some modifications. Ten milligrams of extracted fat were weighed in a 2 mL vial and dissolved in 1 mL of hexane. Three hundred microliters of 2N methanolic potassium-hydroxide were added, the mixture was shaken vigorously for 30 s and allowed to react for a total of 6 min at room temperature (about 20 °C). The hexane phase (1 μL), containing the FAMEs, was analyzed by high-resolution gas chromatography (HRGC). A DANI Master gas-chromatograph (DANI Instruments SpA, Milan, Italy), equipped with a PTV (programmed temperature vaporizer), a split injector, and a flame ionization detector (FID) was used. Analysis was performed with a Quadrex 007-23 column for FAME (60 m, 0.25 mm i.d., 0.25 mm film thickness; Quadrex Corp., Belthany, CT, USA). The carrier gas was high-purity helium with a flow rate of 1.2 mL/min. The injector and detector temperatures were kept at 240 °C. The column oven temperature was programmed at 80 °C for 5 min, from 80 to 165 °C at 5 °C/min for 5 min, from 165 to 230 °C at 3 °C/min for 0.5 min and from 230 to 260 °C at 50 °C/min for 2 min. The FID conditions were a 10:1 ratio of air:hydrogen and a temperature

of 260 °C. The identification and the quantification of separated peaks were performed using the Supelco 37 Component FAME MIX (Supelco Bellofonte, PA, USA) as external standards. The fatty acids were determined using the external standard method and were expressed as percentage of total fatty acids.

### 2.8. Volatile Organic Compounds Analysis

The extraction and analysis of volatile organic compounds (VOC) was performed using SPME-GC/MS, according to Lee et al. [15], with some modifications. The solid-phase microextraction (SPME) device (Supelco Co., Bellefonte, PA, USA) equipped with a 50/30-μm thickness divinyl-benzene/carboxen/polydimethylsiloxane (DVB/CAR/PDMS) fiber coated with 2-cm length stationary phase was used. Twenty-five grams of frozen grated cheese was transferred into a 100 mL bottle, previously added with 25 mL of deionized water, 50 μL of 2-methyl-3-heptanone as internal standard (408 mg L$^{-1}$), and 12.5 g of sodium phosphate (Sigma, St. Louis, MO, USA). Samples were homogenized and heated on a heating magnetic stirrer. Then the SPME device was hermetically put in the bottles containing the samples and left for 30 min at 40 °C. The SPME was introduced directly into the GC injector where the thermal desorption of the analytes was performed at 250 °C for 10 min. A GC system 6890N equipped with a mass detector 5973 (Agilent Technologies, Palo Alto, CA, USA) was used. The VOCs were separated on a 30 m × 0.250 mm capillary column coated with a 0.25 μm film of 5% diphenyl l95% dimethylpolysiloxane (HP5MS J&W Scientific, Folson, CA, USA). Splitless injection was used for the samples. The column oven temperature was programmed at 10 °C/min from an initial temperature of 50 (held for 2 min) to 150 °C, then at 15 °C/min to 300 °C, which was held for 10 min. The injection and ion source temperatures were 250 and 230 °C, respectively. Helium (99.999%) was used as carrier gas at a flow rate of 1 mL/min. The ionizing electron energy was 70 eV and the mass range scanned was 40–450 amu in full-scan acquisition mode. The compounds were identified using the NIST Atomic Spectra Database version1.6 and verified by the retention indices. The VOCs were calculated by the internal standard method and were expressed as μg/kg of cheese.

### 2.9. Statistical Analysis

All analyses were performed in triplicate. Significant differences among the different samples were determined by one-way ANOVA statistical analysis and PCA (Principal Component Analysis). Tukey's test was used to discriminate among the means of the variables. Differences with $p < 0.05$ were considered significant. The data elaboration was carried out using XLStat version 2009.3.02 (Addinsoft Corp., Paris, France).

## 3. Results

### 3.1. Moisture, pH, and Titratable Acidity

Provolone del Monaco's typical shape is shown in Figure 1; the chemical composition at different ripening times and sampled at different portions is reported in Table 1.

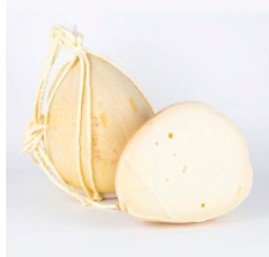

**Figure 1.** Provolone del Monaco.

**Table 1.** Chemical composition of cheese samples (mean values ± SD) during 270 days of ripening.

| Samples | Ripening Days | pH | Moisture % | Titratable Acidity (Lact. ac. %) | Total N (g/100 g dw) | Soluble N (g/100 g dw) | SN/TN*100 | Fat (g/100 g dw) |
|---|---|---|---|---|---|---|---|---|
| C3 | 0 | 5.03 [b] ± 0.25 | 46.20 [a] ± 1.34 | 0.17 ± 0.02 | 7.17 ± 0.15 | 0.49 [c] ± 0.01 | 6.87 [c] ± 0.05 | 42.96 ± 2.81 |
| | 90 | 5.63 [a,b] ± 0.02 | 40.03 [b] ± 0.85 | 0.14 ± 0.01 | 6.82 ± 0.13 | 1.00 [b] ± 0.01 | 14.74 [b] ± 0.15 | 41.65 ± 1.13 |
| | 180 | 5.77 [a] ± 0.18 | 40.61 [b] ± 0.54 | 0.22 + 0.03 | 7.02 ± 0.03 | 1.08 [b] ± 0.03 | 15.36 [b] ± 0.42 | 43.45 ± 0.13 |
| | 270 | 5.68 [a,b] ± 0.10 | 38.35 [b] ± 0.13 | 0.22 ± 0.03 | 7.06 ± 0.18 | 1.70 [a] ± 0.08 | 24.02 [a] ± 0.59 | 39.89 ± 2.09 |
| C5 | 0 | 5.30 [c] ± 0.04 | 47.05 [a] ± 0.92 | 0.16 [a,b] ± 0.01 | 7.10 ± 0.23 | 0.50 [d] ± 0.02 | 6.99 [c] ± 0.58 | 42.84 ± 1.83 |
| | 90 | 5.79 [b] ± 0.03 | 42.93 [b] ± 0.64 | 0.13 [b] ± 0.01 | 7.11 ± 0.08 | 1.12 [c] ± 0.06 | 15.70 [b] ± 0.61 | 42.65 ± 1.26 |
| | 180 | 5.97 [a,b] ± 0.01 | 42.11 [b,c] ± 1.05 | 0.21 [a] ± 0.01 | 7.12 ± 0.24 | 1.30 [b] ± 0.04 | 18.33 [b] ± 1.14 | 43.51 ± 1.17 |
| | 270 | 6.02 [a] ± 0.08 | 39.51 [c] ± 0.27 | 0.22 [a] ± 0.03 | 7.17 ± 0.03 | 2.04 [a] ± 0.06 | 28.46 [a] ± 0.88 | 41.88 ± 1.24 |
| S3 | 0 | 5.23 [b] ± 0.06 | 43.80 [a] ± 1.89 | 0.18 [b] ± 0.01 | 7.01 ± 0.56 | 0.40 [c] ± 0.01 | 5.73 [c] ± 0.27 | 41.67 ± 3.28 |
| | 90 | 5.49 [a,b] ± 0.03 | 31.63 [b] ± 1.18 | 0.13 [c] ± 0.01 | 6.62 ± 0.12 | 0.80 [b] ± 0.01 | 12.11 [b] ± 0.40 | 39.40 ± 2.39 |
| | 180 | 5.70 [a] ± 0.11 | 31.92 [b] ± 3.03 | 0.24 [a] ± 0.01 | 7.15 ± 0.25 | 1.03 [a] ± 0.04 | 14.44 [a] ± 0.06 | 42.10 ± 3.35 |
| | 270 | 5.69 [a] ± 0.06 | 29.35 [b] ± 0.61 | 0.21 [a,b] ± 0.01 | 7.03 ± 0.04 | 1.14 [a] ± 0.06 | 16.16 [a] ± 0.84 | 39.84 ± 0.17 |
| S5 | 0 | 5.29 [c] ± 0.02 | 45.76 [a] ± 0.02 | 0.18 [b] ± 0.01 | 7.53 [a] ± 0.04 | 0.47 [b] ± 0.01 | 6.30 [b] ± 0.12 | 42.91 ± 3.22 |
| | 90 | 5.46 [b] ± 0.01 | 36.89 [b] ± 2.69 | 0.13 [c] ± 0.01 | 6.74 [c] ± 0.04 | 1.29 [a] ± 0.09 | 19.12 [a] ± 1.28 | 41.39 ± 1.86 |
| | 180 | 5.73 [a] ± 0.02 | 34.81 [b] ± 2.33 | 0.22 [a] ± 0.01 | 7.36 [b,c] ± 0.21 | 1.18 [a] ± 0.04 | 16.06 [a] ± 0.11 | 42.18 ± 4.72 |
| | 270 | 5.77 [a] ± 0.08 | 31.69 [b] ± 0.31 | 0.21 [a,b] ± 0.01 | 7.02 [b,c] ± 0.12 | 1.21 [a] ± 0.11 | 17.25 [a] ± 1.34 | 37.18 ± 0.53 |

a–c: Different letters in the same column correspond to significant differences ($p < 0.05$); C3 core portion of 3 kg size; C5 core portion of 5 kg size; S3 under rind portion of 3 kg size; S5 under rind portion of 5 kg size.

The cheese at time 0 showed a water content ranging from 43.80 to 47.05%. In particular, the moisture percentage was 46.20% and 47.05% in the core (C) portions and 43.80 and 45.76% in the portion under the rind (S) portions, in the 3-kg and 5-kg sized cheeses, respectively. At 90 days of ripening, the highest loss of moisture was observed. In fact, the values decreased to 40.03% and 42.93% in C portions and to 31.63% and 36.89% in S portions in the 3-kg and 5 kg-sized cheeses, respectively. After 90 days of ripening, the loss of water was less marked, probably due to the definitive rind formation. The S portions of both sizes had lower moisture contents than the C portions because the S portion was located closer to the external surface of the cheese in comparison with the C portions, producing a faster loss of water.

The pH values detected in the analyzed samples ranged from 5.03 to 6.02 in the C portions and from 5.23 to 5.77 in the S portions in the 3-kg and 5 kg-sized cheeses, respectively. The increase of pH in cheese during ripening has been associated with the proteolytic process that released large amounts of nitrogenated alkaline compounds [16]. After 270 days of ripening, the 5-kg sized cheese showed slightly higher pH values than the 3-kg sized cheese, probably because of the more abundant water content in the 5-kg sized cheese [17].

With reference to the titratable acidity, the data obtained showed values ranging from 0.17 to 0.22 of lactic acid/100 g of cheese in the C portions and from 0.19 to 0.21 g of lactic acid/100 g of cheese in the S portions during 270 days of ripening in the 3-kg and 5-kg sized cheeses, respectively. In particular, during the first 90 days of storage, all the samples presented a decrease of titratable acidity in comparison with the data observed in the first stage of the maturation period. A significant increase in this parameter was observed at 270 days, possibly due to lactic acid formation from residual lactose still present in the cheese [18].

## 3.2. Nitrogen Content

The total nitrogen (TN) content values measured in cheese samples ranged from 6.82 to 7.17% dw in the C portions and from 6.62 to 7.53% dw in the S portions in the 3-kg and 5-kg sized cheeses, respectively (Table 1). No significant increase was observed during the storage process in all examined samples. The S samples showed higher TN contents in comparison with the C samples, probably due to the loss of water and to the dry matter (DM) concentration [3].

Regarding soluble nitrogen (SN) at pH 4.6, at time 0, the data analyzed ranged from 0.49 to 0.50% in the C portions and from 0.40 to 0.47% in the S portions in the 3-kg and 5-kg sized cheeses, respectively. A significant increase was observed in all samples at 90 days of ripening, with values ranging from 1.00 to 1.12% in the C portions and from 0.80 to 1.29% in the S portions in both cheese sizes (3 kg and 5 kg, respectively). This increment of the parameter studied can be explained by the fact that proteolysis is directly correlated with the increase in pH during the ripening phase [19]. After 270 days of ripening, the C portions showed SN values higher than the S portions (1.70% and 2.04% with respect to 1.14% and 1.21%, in 3-kg and 5-kg sized cheeses, respectively), according to Gobbetti et al. [20]. This difference is probably due to their water contents; in fact, the moisture content of the substrate can determine a minor proteolytic activity [13]. Considering that proteolysis is one of the principal biochemical transformations produced during the ripening of cheese, the ratio between soluble nitrogen at pH 4.6 and total nitrogen (SN/TN) was calculated. These data, called the maturation index, represent a good indicator of cheese ripening and of its protolithic activity.

In particular, at the beginning of the ripening period, the SN/TN data detected ranged from 6.87 to 6.99 for the C samples and from 5.73 to 6.30 for the S samples, in the 3 kg and 5 kg cheeses, respectively. These data drastically increased during the storage period, particularly at 90 and 180 days of ripening, with data ranging from 15.36 to 18.33 for the C samples and from 14.44 to 16.06 for the S samples in the 3-kg and 5-kg sized cheeses, respectively. After 270 days of ripening, the C portions of both 3 kg and 5 kg samples showed SN/TN values higher than those of the S portions because of the more abundant water content in the C portions [17]. In particular, the SN/TN data detected in the C portions were 24.02 and 28.46 for 3-kg and 5-kg sized cheeses, respectively, whereas in the S portion, they were 16.16

and 17.25 for 3-kg and 5-kg sized cheeses, respectively, indicating that Provolone del Monaco PDO cheese is a firm/hard pasta filata cheese [21].

### 3.3. Fat Content

One of the most important analytical parameters that allow Provolone del Monaco cheese to obtain the PDO brand is the fat content on dry matter (F/DM). No significant differences were observed compared with the time zero data, even if after 6 months of ripening, the fat content (g/100 g dw) detected was higher than 42% and ranged from 42.10 to 43.51%. At 270 days ripening, cheese samples showed fat content (g/100 g dw) mean values of 39.86% and 39.53% for 3 kg and 5 kg cheese, respectively.

### 3.4. Fatty Acid Profile

Twenty-two fatty acids in concentration >0.1% were identified in Provolone del Monaco cheese samples, from C4:0 (butyric acid) to C20:3n3 (eicosatrienoic acid), including c9t11-conjugated linoleic acid isomer (CLAc9t11). The identification of the fatty acids is reported in Table 2a,b.

The fatty acid profile of cheese largely reflected that of the raw milk from which cheese was made [22–25]. Medium chain fatty acids (MCFAs, from C10:0 to C16:1) were the most abundant in all the samples, with values reaching 49%. Among MCFAs, palmitic acid (C16:0) was the most representative, with a percentage of approximately 29%. Long chain fatty acids (LCFAs, from C17:0 to C20:3n3) reached approximately 44%, with oleic acid (C18:1n9c) as the most abundant (21%). Short chain fatty acids (SCFAs, from C4:0 to C8:0) reached values of approximately 6%. Concentrations of approximately 0.65% of c9t11 CLA were detected. In recent years, attention has been paid towards the potential benefits of short chain fatty acids and CLA [26,27]. For instance, the roles and importance of butyric acid has been expanding steadily: immunomodulatory activity of butyric acid in the gastrointestinal tract, the positive effects on irritable bowel syndrome, in non-specific inflammatory bowel diseases, in cancer prevention and treatment [28,29].

The analysis revealed that saturated fatty acids ($\Sigma$SFAss) represented approximately 69%, with C16:0 as the most abundant $\Sigma$SFA. Among unsaturated fatty acids ($\Sigma$UFAs), monounsaturated ($\Sigma$MUFA) represented approximately 24% of $\Sigma$UFA, and the most abundantly detected $\Sigma$MUFA was oleic acid (C18:1n9c). Finally, polyunsaturated fatty acids ($\Sigma$PUFA) represented just 6%, with linoleic acid (C18:2n6c) as the most representative $\Sigma$PUFA (3.6%). All these results are in agreement with Romano et al. [3]. A few significant differences in fatty acid concentrations were detected during ripening. Regarding the 3 kg cheese samples, capric acid (C10:0) showed a significant increase (from 2.49% to 2.83%) in the C portions; the linolelaidic acid value (C18:2n6t), however, decreased significantly (from 0.57% to 0.53%) in the S portions. In the 5 kg samples, caproic acid (C6:0) showed a significant increase (from 1.90% to 2.21%) in the S portions; oleic acid (C18:1n9c), on the opposite, showed a significant decrease (from 21.19% to 20.77%) in the S portions.

The reduction of the LCFA and the increase of the SCFA observed could be due to the action of indigenous milk lipases [6] or to the action of gastric lipases present in the goat rennet used to produce Provolone del Monaco cheese [3]. These lipases preferentially hydrolyze long chain fatty acids to produce short chain fatty acids. Depending on their concentration and perception threshold, volatile fatty acids can either positively or negatively contribute to the aroma of the cheese or to a rancidity defect [8].

Lipolysis is classified as spontaneous or induced [30]. Free fatty acid flavors are produced by lipase activity and originate from short-chain FFA, with described flavors that are reminiscent of vinegar, cheese, sweat, and soap [31]. These FFA are generated by the enzymatic hydrolysis of ester bonds of triglycerides. Oxidized flavors are due to the autoxidation of fatty acids and are characterized by cardboard, metallic, and mushroom flavors [32,33].

**Table 2.** (**a**) Composition of fatty acids (% ± SD) in 3-kg size samples during 270 days of ripening. (**b**) Composition of fatty acids (% ± SD) in 5-kg size samples during 270 days of ripening.

| | C | | | | S | | | |
|---|---|---|---|---|---|---|---|---|
| | 0 | 90 | 180 | 270 | 0 | 90 | 180 | 270 |
| | | | | (a) | | | | |
| C4:0 | 3.04 ± 0.05 | 3.07 ± 0.00 | 3.13 ± 0.02 | 3.10 ± 0.04 | 2.81 ± 0.18 | 3.00 ± 0.18 | 3.01 ± 0.28 | 3.22 ± 0.22 |
| C6:0 | 1.93 [a,b] ± 0.05 | 1.84 [b] ± 0.05 | 2.03 [a,b] ± 0.05 | 2.13 [a,b] ± 0.07 | 1.93 [a,b] ± 0.07 | 1.97 [a,b] ± 0.09 | 2.05 [a,b] ± 0.11 | 2.19 [a] ± 0.13 |
| C8:0 | 1.18 ± 0.10 | 1.14 ± 0.00 | 1.23 ± 0.02 | 1.42 ± 0.24 | 1.18 ± 0.08 | 1.27 ± 0.08 | 1.20 ± 0.12 | 1.26 ± 0.13 |
| C10:0 | 2.49 [c] ± 0.04 | 2.51 [b,c] ± 0.01 | 2.78 [a,b] ± 0.05 | 2.83 [a] ± 0.07 | 2.65 [a,b,c] ± 0.08 | 2.63 [a,b,c] ± 0.02 | 2.55 [a,b,c] ± 0.11 | 2.53 [b,c] ± 0.12 |
| C12:0 | 2.94 ± 0.01 | 2.88 ± 0.01 | 2.97 ± 0.08 | 3.00 ± 0.05 | 3.00 ± 0.06 | 2.98 ± 0.03 | 2.93 ± 0.07 | 2.88 ± 0.07 |
| C14:0 | 10.90 [b] ± 0.16 | 10.94 [a,b] ± 0.01 | 11.23 [a] ± 0.06 | 10.77 [b] ± 0.04 | 11.04 [a,b] ± 0.08 | 11.01 [a,b] ± 0.09 | 10.97 [a,b] ± 0.02 | 10.75 [b] ± 0.04 |
| C14:1 | 0.95 ± 0.00 | 0.94 ± 0.00 | 0.96 ± 0.01 | 0.96 ± 0.01 | 0.97 ± 0.02 | 0.97 ± 0.00 | 0.96 ± 0.01 | 0.95 ± 0.01 |
| C15:0 | 1.07 [b] ± 0.02 | 1.10 [a,b] ± 0.01 | 1.08 [a,b] ± 0.00 | 1.09 [a,b] ± 0.01 | 1.12 [a] ± 0.00 | 1.12 [a,b] ± 0.00 | 1.11 [a,b] ± 0.01 | 1.11 [a,b] ± 0.02 |
| C16:0 | 29.52 ± 0.10 | 29.63 ± 0.03 | 29.28 ± 0.07 | 28.93 ± 0.05 | 29.34 ± 0.29 | 29.31 ± 0.05 | 29.31 ± 0.39 | 29.21 ± 0.36 |
| C16:1 | 1.49 [b,c] ± 0.02 | 1.48 [c] ± 0.03 | 1.48 [c] ± 0.01 | 1.53 [a,b,c] ± 0.01 | 1.55 [a,b] ± 0.02 | 1.51 [a,b,c] ± 0.00 | 1.52 [a,b,c] ± 0.01 | 1.56 [a] ± 0.00 |
| C17:0 | 0.89 [b] ± 0.01 | 0.94 [a,b] ± 0.01 | 0.92 [a,b] ± 0.02 | 0.95 [a,b] ± 0.04 | 0.97 [a,b] ± 0.04 | 0.98 [a] ± 0.00 | 0.99 [a] ± 0.03 | 1.01 [a] ± 0.02 |
| C17:1 | 0.22 [d] ± 0.00 | 0.23 [c,d] ± 0.00 | 0.23 [b,c,d] ± 0.01 | 0.25 [a,b,c] ± 0.01 | 0.25 [a,b,c] ± 0.01 | 0.25 [a,b] ± 0.00 | 0.25 [a] ± 0.00 | 0.25 [a,b] ± 0.00 |
| C18:0 | 15.26 ± 0.07 | 15.26 ± 0.04 | 14.96 ± 0.10 | 15.05 ± 0.10 | 15.07 ± 0.17 | 15.13 ± 0.01 | 15.08 ± 0.17 | 15.16 ± 0.22 |
| C18:1n9t | 0.45 ± 0.03 | 0.48 ± 0.04 | 0.42 ± 0.01 | 0.47 ± 0.04 | 0.45 ± 0.01 | 0.48 ± 0.01 | 0.45 ± 0.04 | 0.51 ± 0.02 |
| C18:1n9c | 21.24 [a] ± 0.11 | 21.18 [a,b] ± 0.01 | 20.96 [b,c] ± 0.02 | 21.03 [a,b,c] ± 0.09 | 20.99 [b,c] ± 0.05 | 20.82 [c] ± 0.01 | 20.96 [b,c] ± 0.08 | 20.86 [c] ± 0.01 |
| C18:2n6t | 0.52 [b] ± 0.01 | 0.51 [b] ± 0.00 | 0.51 [b] ± 0.01 | 0.52 [b] ± 0.00 | 0.57 [a] ± 0.02 | 0.54 [a,b] ± 0.00 | 0.54 [a,b] ± 0.00 | 0.53 [b] ± 0.01 |
| C18:2n6c | 3.62 [a] ± 0.03 | 3.55 [a,b] ± 0.01 | 3.52 [b] ± 0.04 | 3.59 [a,b] ± 0.02 | 3.60 [a] ± 0.02 | 3.61 [a] ± 0.00 | 3.61 [a] ± 0.00 | 3.58 [a,b] ± 0.01 |
| C20:0 | 0.21 ± 0.01 | 0.21 ± 0.01 | 0.20 ± 0.01 | 0.22 ± 0.01 | 0.22 ± 0.00 | 0.13 ± 0.00 | 0.23 ± 0.00 | 0.23 ± 0.00 |
| C18:3n3 | 0.63 ± 0.02 | 0.62 ± 0.00 | 0.63 ± 0.02 | 0.65 ± 0.02 | 0.66 ± 0.01 | 0.67 ± 0.00 | 0.67 ± 0.01 | 0.66 ± 0.00 |
| C9t11 | 0.64 [a,b] ± 0.00 | 0.63 [a,b] ± 0.00 | 0.61 [b] ± 0.01 | 0.65 [a] ± 0.02 | 0.65 [a] ± 0.00 | 0.65 [a] ± 0.00 | 0.65 [a] ± 0.00 | 0.66 [a] ± 0.01 |
| C20:3n6 | 0.17 [c] ± 0.00 | 0.17 [c] ± 0.00 | 0.17 [b,c] ± 0.00 | 0.17 [b,c] ± 0.00 | 0.19 [a,b] ± 0.01 | 0.19 [a] ± 0.00 | 0.19 [a] ± 0.00 | 0.18 [a,b,c] ± 0.00 |
| C20:3n3 | 0.10 [c] ± 0.00 | 0.10 [b,c] ± 0.00 | 0.10 [b,c] ± 0.01 | 0.11 [a,b,c] ± 0.00 | 0.11 [a,b,c] ± 0.00 | 0.11 [a,b,c] ± 0.00 | 0.11 [a,b] ± 0.00 | 0.12 [a] ± 0.00 |
| SFAs | 69.76 ± 0.14 | 69.94 ± 0.07 | 70.21 ± 0.14 | 69.87 ± 0.21 | 69.77 ± 0.06 | 69.94 ± 0.03 | 69.84 ± 0.10 | 69.91 ± 0.02 |
| MUFAs | 24.45 ± 0.10 | 24.38 ± 0.05 | 24.14 ± 0.06 | 24.33 ± 0.15 | 24.32 ± 0.06 | 24.15 ± 0.01 | 24.25 ± 0.11 | 24.23 ± 0.01 |
| PUFAs | 5.77 [a,b,c] ± 0.04 | 5.66 [b,c] ± 0.02 | 5.63 [c] ± 0.09 | 5.79 [a,b,c] ± 0.06 | 5.88 [a] ± 0.01 | 5.87 [a] ± 0.01 | 5.86 [a] ± 0.01 | 5.82 [a,b] ± 0.01 |
| UFAs | 30.21 ± 0.15 | 30.04 ± 0.07 | 29.77 ± 0.14 | 30.12 ± 0.21 | 30.20 ± 0.05 | 30.02 ± 0.02 | 30.12 ± 0.10 | 30.05 ± 0.02 |
| n3 | 0.75 ± 0.02 | 0.74 ± 0.01 | 0.74 ± 0.03 | 0.78 ± 0.03 | 0.78 ± 0.01 | 0.80 ± 0.01 | 0.80 ± 0.00 | 0.79 ± 0.00 |
| n6 | 4.38 [a,b,c] ± 0.02 | 4.30 [c,d] ± 0.01 | 4.27 [d] ± 0.05 | 4.35 [b,c,d] ± 0.02 | 4.45 [a] ± 0.00 | 4.42 [a,b] ± 0.00 | 4.42 [a,b] ± 0.01 | 4.37 [a,b,c] ± 0.00 |
| n3/n6 | 0.17 ± 0.00 | 0.17 ± 0.00 | 0.17 ± 0.01 | 0.18 ± 0.01 | 0.18 ± 0.00 | 0.18 ± 0.00 | 0.18 ± 0.00 | 0.18 ± 0.00 |
| SCFAs (4–8) | 6.15 ± 0.20 | 6.06 ± 0.04 | 6.39 ± 0.01 | 6.65 ± 0.35 | 5.92 ± 0.32 | 6.24 ± 0.35 | 6.25 ± 0.51 | 6.67 ± 0.49 |
| MCFs (10–16) | 49.49 [a,b] ± 0.24 | 49.58 [a,b] ± 0.05 | 49.90 [a] ± 0.26 | 49.23 [a,b] ± 0.02 | 49.81 [a,b] ± 0.06 | 49.66 [a,b] ± 0.18 | 49.48 [a,b] ± 0.19 | 49.11 [b] ± 0.22 |
| LCFAs (17–24) | 44.36 ± 0.03 | 44.36 ± 0.00 | 43.71 ± 0.25 | 44.12 ± 0.37 | 44.27 ± 0.26 | 44.10 ± 0.16 | 44.27 ± 0.32 | 44.23 ± 0.26 |

**Table 2.** *Cont.*

| | C | | | | S | | | |
|---|---|---|---|---|---|---|---|---|
| | **0** | **90** | **180** | **270** | **0** | **90** | **180** | **270** |
| | (b) | | | | | | | |
| **C4:0** | 2.81 a,b ± 0.08 | 2.72 a,b ± 0.24 | 2.75 a,b ± 0.25 | 2.77 a,b ± 0.08 | 2.84 a,b ± 0.12 | 2.62 b ± 0.13 | 2.99 a,b ± 0.21 | 3.32 a ± 0.19 |
| **C6:0** | 1.84 b ± 0.04 | 1.90 a,b ± 0.05 | 1.87 b ± 0.07 | 1.85 b ± 0.06 | 1.90 b ± 0.04 | 1.76 b ± 0.06 | 2.00 a,b ± 0.16 | 2.21 a ± 0.05 |
| **C8:0** | 1.13 ± 0.02 | 1.15 ± 0.09 | 1.18 ± 0.04 | 1.15 ± 0.02 | 1.16 ± 0.00 | 1.13 ± 0.02 | 1.16 ± 0.04 | 1.30 ± 0.08 |
| **C10:0** | 2.52 ± 0.03 | 2.46 ± 0.07 | 2.58 ± 0.01 | 2.51 ± 0.01 | 2.49 ± 0.00 | 2.50 ± 0.01 | 2.54 ± 0.10 | 2.52 ± 0.02 |
| **C12:0** | 2.89 ± 0.04 | 2.85 ± 0.02 | 2.91 ± 0.03 | 2.89 ± 0.02 | 2.86 ± 0.00 | 2.89 ± 0.01 | 2.94 ± 0.12 | 3.08 ± 0.13 |
| **C14:0** | 10.93 ± 0.01 | 10.86 ± 0.00 | 10.93 ± 0.08 | 10.92 ± 0.06 | 10.89 ± 0.02 | 10.96 ± 0.04 | 10.99 ± 0.15 | 10.94 ± 0.04 |
| **C14:1** | 0.95 ± 0.00 | 0.95 ± 0.01 | 0.97 ± 0.00 | 0.97 ± 0.00 | 0.95 ± 0.00 | 0.97 ± 0.00 | 0.97 ± 0.02 | 0.98 ± 0.02 |
| **C15:0** | 1.10 ± 0.01 | 1.12 ± 0.00 | 1.11 ± 0.00 | 1.12 ± 0.00 | 1.11 ± 0.00 | 1.13 ± 0.01 | 1.12 ± 0.01 | 1.10 ± 0.03 |
| **C16:0** | 29.40 ± 0.13 | 29.33 ± 0.23 | 29.29 ± 0.09 | 29.34 ± 0.14 | 29.37 ± 0.16 | 29.44 ± 0.05 | 29.19 ± 0.22 | 29.11 ± 0.07 |
| **C16:1** | 1.53 a,b ± 0.01 | 1.58 a ± 0.02 | 1.54 a,b ± 0.01 | 1.55 a,b ± 0.00 | 1.53 a,b ± 0.03 | 1.59 a ± 0.01 | 1.54 a,b ± 0.01 | 1.52 b ± 0.01 |
| **C17:0** | 0.97 ± 0.01 | 0.99 ± 0.01 | 0.98 ± 0.00 | 0.99 ± 0.00 | 1.00 ± 0.01 | 1.01 ± 0.02 | 0.99 ± 0.04 | 0.94 ± 0.05 |
| **C17:1** | 0.25 ± 0.00 | 0.26 ± 0.00 | 0.25 ± 0.00 | 0.25 ± 0.00 | 0.27 ± 0.03 | 0.26 ± 0.00 | 0.25 ± 0.00 | 0.25 ± 0.01 |
| **C18:0** | 15.31 a,b ± 0.09 | 15.32 a,b ± 0.01 | 15.16 a,b ± 0.02 | 15.28 a,b ± 0.06 | 15.32 a,b ± 0.08 | 15.46 a ± 0.03 | 15.16 a,b ± 0.25 | 14.96 b ± 0.15 |
| **C18:1n9t** | 0.49 ± 0.00 | 0.52 ± 0.00 | 0.46 ± 0.00 | 0.51 ± 0.02 | 0.51 ± 0.00 | 0.50 ± 0.00 | 0.49 ± 0.05 | 0.45 ± 0.05 |
| **C18:1n9c** | 21.30 a,b ± 0.12 | 21.29 a,b ± 0.13 | 21.39 a ± 0.13 | 21.19 a,b ± 0.09 | 21.19 a,b ± 0.01 | 21.00 a,b,c ± 0.03 | 20.96 b,c ± 0.16 | 20.77 c ± 0.00 |
| **C18:2n6t** | 0.52 ± 0.00 | 0.55 ± 0.01 | 0.52 ± 0.01 | 0.53 ± 0.01 | 0.54 ± 0.01 | 0.54 ± 0.00 | 0.55 ± 0.03 | 0.52 ± 0.02 |
| **C18:2n6c** | 3.63 a,b ± 0.02 | 3.67 a ± 0.05 | 3.64 a,b ± 0.01 | 3.69 a ± 0.02 | 3.61 a,b ± 0.00 | 3.65 a,b ± 0.01 | 3.58 b ± 0.02 | 3.61 a,b ± 0.01 |
| **C20:0** | 0.22 ± 0.00 | 0.24 ± 0.00 | 0.22 ± 0.00 | 0.23 ± 0.00 | 0.23 ± 0.00 | 0.24 ± 0.01 | 0.23 ± 0.01 | 0.22 ± 0.02 |
| **C18:3n3** | 0.66 ± 0.00 | 0.68 ± 0.01 | 0.66 ± 0.00 | 0.68 ± 0.01 | 0.66 ± 0.00 | 0.68 ± 0.01 | 0.66 ± 0.01 | 0.65 ± 0.02 |
| **C9t11** | 0.66 ± 0.00 | 0.67 ± 0.01 | 0.66 ± 0.01 | 0.66 ± 0.00 | 0.66 ± 0.01 | 0.67 ± 0.00 | 0.66 ± 0.01 | 0.64 ± 0.01 |
| **C20:3n6** | 0.18 b ± 0.00 | 0.19 b ± 0.00 | 0.18 b ± 0.00 | 0.19 b ± 0.00 | 0.18 b ± 0.00 | 0.27 a ± 0.00 | 0.18 b ± 0.00 | 0.18 b ± 0.00 |
| **C20:3n3** | 0.11 ± 0.00 | 0.12 ± 0.00 | 0.11 ± 0.00 | 0.12 ± 0.00 | 0.11 ± 0.00 | 0.12 ± 0.00 | 0.11 ± 0.00 | 0.11 ± 0.00 |
| **SFAs** | 69.48 a,b ± 0.17 | 69.29 b ± 0.22 | 69.39 b ± 0.12 | 69.45 a,b ± 0.13 | 69.56 a,b ± 0.08 | 69.52 a,b ± 0.05 | 69.77 a,b ± 0.27 | 70.08 a ± 0.12 |
| **MUFAs** | 24.62 a ± 0.13 | 24.70 a ± 0.14 | 24.70 a ± 0.11 | 24.56 a ± 0.09 | 24.55 a ± 0.06 | 24.41 a,b ± 0.04 | 24.35 a,b ± 0.18 | 24.07 b ± 0.05 |
| **PUFAs** | 5.86 a,b ± 0.03 | 5.97 a,b ± 0.08 | 5.87 a,b ± 0.01 | 5.95 a,b ± 0.03 | 5.85 a,b ± 0.01 | 6.03 a ± 0.01 | 5.84 a,b ± 0.09 | 5.81 b ± 0.07 |
| **UFAs** | 30.48 a,b ± 0.17 | 30.67 a ± 0.22 | 30.57 a ± 0.12 | 30.51 a ± 0.13 | 30.40 a,b ± 0.07 | 30.44 a,b ± 0.05 | 30.19 a,b ± 0.27 | 29.88 b ± 0.12 |
| **n3** | 0.79 ± 0.00 | 0.82 ± 0.01 | 0.79 ± 0.00 | 0.81 ± 0.01 | 0.79 ± 0.01 | 0.82 ± 0.01 | 0.79 ± 0.02 | 0.78 ± 0.02 |
| **n6** | 4.41 a,b ± 0.03 | 4.49 a,b ± 0.06 | 4.42 a,b ± 0.01 | 4.48 a,b ± 0.03 | 4.40 b ± 0.00 | 4.54 a ± 0.02 | 4.39 b ± 0.06 | 4.39 b ± 0.03 |
| **n3/n6** | 0.18 ± 0.00 | 0.18 ± 0.00 | 0.18 ± 0.00 | 0.18 ± 0.00 | 0.18 ± 0.00 | 0.18 ± 0.00 | 0.18 ± 0.00 | 0.18 ± 0.00 |
| **SCFAs(4–8)** | 5.78 a,b ± 0.01 | 5.78 a,b ± 0.38 | 5.80 a,b ± 0.35 | 5.77 a,b ± 0.15 | 5.89 a,b ± 0.16 | 5.50 b ± 0.21 | 6.16 a,b ± 0.41 | 6.83 a ± 0.32 |
| **MCFAs (10–16)** | 49.43 ± 0.04 | 49.28 ± 0.16 | 49.45 ± 0.18 | 49.42 ± 0.22 | 49.33 ± 0.16 | 49.60 ± 0.11 | 49.42 ± 0.17 | 49.39 ± 0.08 |
| **LCFAs (17–24)** | 44.78 a,b ± 0.03 | 44.94 a ± 0.23 | 44.75 a,b ± 0.17 | 44.81 a,b ± 0.07 | 44.78 a,b ± 0.00 | 44.89 a ± 0.10 | 44.42 a,b ± 0.58 | 43.78 b ± 0.40 |

a–c: Different letters in the same row correspond to significant differences ($p < 0.05$); *C* core portion, *S* under rind portion; SFAs: saturated fatty acids; MUFAs: monounsaturated fatty acids; PUFAs: polyunsaturated fatty acids; UFAs: unsaturated fatty acids; SCFAs: short-chain fatty acids; MCFAs: medium-chain fatty acids; LCFAs: long-chain fatty acids.

### 3.5. Volatile Organic Compounds

Volatile organic compounds (VOCs) identified in the Provolone del Monaco cheese samples were divided into 6 families, including organic acids, alcohols, ketones, aldehydes, esters, and other compounds (Table 3).

Organic acids were the most abundant VOCs detected. In particular, acetic acid (ranging from 18.25 to 2503.02 µg/kg), butanoic acid (ranging from 186.02 to 11,469.36 µg/kg), and hexanoic acid (ranging from 203.99 to 5022.33 µg/kg) were found. These molecules are typical of dairy products and come from the oxidative decarboxylation of $\alpha$-keto acids, which originate from branched amino acids released during proteolysis [34]. Organic acids (and relative quantities) are the main responsibles for the Provolone del Monaco PDO cheese aroma, giving it its pungent, sour, cheesy, and buttery odors [35–38].

Alcohols and ketones are the second and third most abundant families detected, respectively, and arise from β-oxidation and decarboxylation of free fatty acids released during lipolysis [34]. Most likely, for this reason, no alcohols were revealed at the beginning of ripening. Among the alcohols, ethanol (ranging from 9.49 to 56.73 µg/kg), 1-butanol (ranging from 6.38 to 79.81 µg/kg), and 2-ethyl-1-hexanol (ranging from 8.20 to 80.20 µg/kg), mostly contribute to the Provolone del Monaco aroma, with characteristic alcoholic, fruity, and spicy aromas, respectively. Among ketones, acetoin (3-hydroxy-2-butanone, ranging from 3.77 to 61.91 µg/kg), which is a characteristic dairy product volatile compound, can be found only in the first days of ripening. Aldehydes (hexanal and nonanal) were found in small amounts during ripening. These compounds are produced from $\alpha$-keto acids by decarboxylation [34] and can be quickly converted into alcohols or their respective acids [39] during ripening. This could explain why hexanal (ranging from 26.60 to 63.66 µg/kg) was found only at time 0 in the cheese samples. Esters were the less abundant family detected. Two molecules were identified, ethyl hexanoate (ranging from 35.24 to 123.33 µg/kg) and butyl butanoate (ranging from 33.92 to 47.24 µg/kg). Their origin could be attributed to the reaction between free fatty acids and alcohols [34].

An integrated and multidisciplinary system of analysis is becoming a valuable tool to analyze and study food, taking into account the various aspects of food quality and composition by employing innovative and emerging technologies with statistical analysis. Such a multi-technique approach to food research enables utilizing all the resources of quality, safety, and traceability in food systems. The joined approach of multidisciplinary analysis techniques of food matrices with the multivariate statistical approach enlarges the possibilities to exploit a wide range of food traits and aspects. The "Integrated Approach" is the key to modern food research and the innovative challenge for analyzing and modelling agro-food systems in their totality [40–43]. To better understand the VOC distribution during ripening, a Principal Components Analysis (PCA) was carried out using the recorded dataset, which is shown in Figure 2.

The PCA illustrates which families are characteristic at different ripening times. The F1 and F2 factors explain 97.8% of the total variability. In particular, the factor F1 explains 74.7%. The main contribution to F1 comes from alcohols (21.3%). The factor F2 explains 23.1% of the variability, with the main contribution given from esters (39.0%), followed by aldehydes (38.7%). As expected, the majority of molecules were found in the last months of ripening. In particular, after 90 and 180 days, cheese samples showed a large amount of organic acids and alcohols. Furthermore, cheese at the 270-day timepoint was characterized mainly by esters, ketones, and other compounds.

**Table 3.** Volatile organic compounds families (µg/kg ± SD) in cheese samples during 270 days of ripening.

| Samples | Ripening Days | Acids | Alcohols | Ketones | Aldehydes | Esters | Others |
|---------|---------------|-------|----------|---------|-----------|--------|--------|
| **C3** | **0** | 642.86 [c] ± 163.29 | 28.37 [c] ± 11.47 | 21.28 [a] ± 0.19 | 34.08 [a] ± 7.34 | n.d. | 12.88 [b,c] ± 0.97 |
| | **90** | 2658.35 [b,c] ± 502.10 | 49.53 [c] ± 3.88 | 3.77 [c] ± 0.06 | n.d. | n.d. | 7.44 [c] ± 1.29 |
| | **180** | 7105.93 [a] ± 2043.31 | 195.40 [a] ± 0.60 | 14.25 [a,b] ± 3.54 | n.d. | n.d. | 25.77 [a,b] ± 5.50 |
| | **270** | 5029.01 [a,b] ± 256.45 | 110.03 [b] ± 18.95 | 6.50 [b,c] ± 2.55 | 21.26 [a] ± 0.88 | 68.87 [a] ± 4.01 | 35.99 [a] ± 2.99 |
| **C5** | **0** | 529.72 [c] ± 54.28 | 42.47 [b] ± 5.77 | 26.00 [a] ± 1.17 | 39.84 [a] ± 3.17 | n.d. | 9.38 [b] ± 1.79 |
| | **90** | 9822.30 [a] ± 327.61 | 384.37 [a] ± 56.89 | 6.81 [b] ± 0.52 | n.d. | 46.78 [a,b] ± 12.61 | 26.48 [a] ± 7.10 |
| | **180** | 6459.75 [b] ± 290.65 | 279.10 [a] ± 43.78 | 10.03 [b] ± 1.58 | n.d. | 47.24 [a,b] ± 7.57 | 14.41 [a,b] ± 0.34 |
| | **270** | 5172.55 [b] ± 915.88 | 343.78 [a] ± 54.10 | 9.41 [b] ± 4.08 | 12.64 [b] ± 2.97 | 69.17 [a] ± 22.67 | 27.94 [a] ± 2.83 |
| **S3** | **0** | 1659.50 [b] ± 34.48 | 22.72 [c] ± 2.33 | 46.96 [b] ± 6.17 | 74.28 [a] ± 4.15 | n.d. | 22.30 ± 7.87 |
| | **90** | 9171.68 [a,b] ± 41.58 | 287.42 [a,b] ± 18.37 | 251.86 [b] ± 5.08 | n.d. | n.d. | 39.45 ± 15.89 |
| | **180** | 4147.72 [a,b] ± 526.24 | 154.72 [b,c] ± 10.36 | 76.98 [b] ± 12.68 | n.d. | n.d. | 14.45 ± 1.95 |
| | **270** | 16,051.11 [a] ± 6335.85 | 360.26 [a] ± 74.44 | 604.81 [a] ± 110.20 | 50.30 [b] ± 0.26 | 123.33 [a] ± 0.41 | 68.57 ± 20.44 |
| **S5** | **0** | 441.49 [b] ± 141.81 | 21.38 [b] ± 4.69 | 26.47 ± 1.97 | 34.74 [a] ± 2.80 | n.d. | 12.67 b ± 0.08 |
| | **90** | 17,539.56 [a] ± 2130.94 | 360.20 [a] ± 88.95 | 511.24 ± 253.21 | n.d. | n.d. | 49.49 [a] ± 6.08 |
| | **180** | 15,607.76 [a] ± 2303.45 | 271.80 [a] ± 28.88 | 219.30 ± 13.33 | 17.50 [a,b] ± 2.15 | n.d. | 44.82 [a,b] ± 6.04 |
| | **270** | 7128.88 [b] ± 2002.45 | 209.47 [a,b] ± 45.13 | 255.68 ± 67.22 | 22.09 [a] ± 9.31 | 62.71 [a] ± 5.76 | 26.57 [a,b] ± 13.52 |

a–c: Different letters in the same column correspond to significant differences ($p < 0.05$); C3 core portion of 3 kg size; C5 core portion of 5 kg size; S3 under rind portion of 3 kg size; S5 under rind portion of 5 kg size; n.d. = not detected.

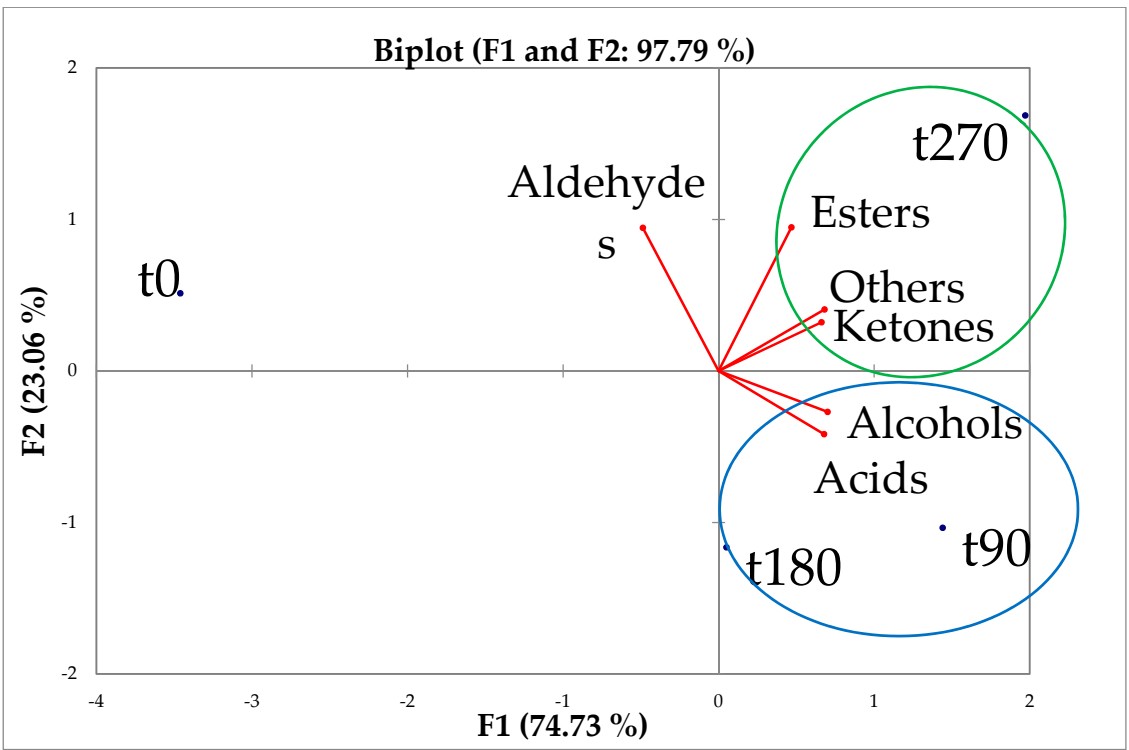

**Figure 2.** Principal components analysis (PCA) of volatile organic compounds (VOC) evaluated in Provolone del Monaco samples after 0 (t0), 90 (t90), 180 (t180), and 270 (t270) days of ripening.

## 4. Conclusions

Investigation and valorization of traditional products is essential for the optimization of their potential beneficial and healthy properties, for the preservation of agro-biodiversity, and sustainability promotion. According to this perspective view, our research supports quality in food and in the meantime helps to promote sustainable resource management through environmental sound farming systems linked to territorial characterization and to local cultural heritage. Provolone del Monaco PDO, an example of traditional cheese, showed important chemical changes during 270 days of ripening. As expected, the main significant differences involved the loss of water (ranging from 47% to 29%), the increase of acidity (from 0.16% to 0.22%), and the large production of nitrogen (from 3.8% to 5%). Cheese samples showed significant differences in maturation index (SN/TN) values (from 6% to 28.5%), indicating that a strong proteolytic activity occurred during ripening. No significant differences were found in the fatty acid profile. Provolone del Monaco cheese is mainly characterized by medium chain saturated fatty acids, with palmitic acid (C16:0) as the most abundant one (29%). Only a few significant differences in the fatty acid profile were found during the 270 days of ripening. With reference to the aromatic profile, the analysis of the data showed that acids are the principal volatile compounds contributing to the cheese aroma with pungent and buttery odors. PCA also showed that at 180 days of ripening, acids, alcohols and ketones represent the most characteristic volatile compound families, and at 270 days, mainly esters can be found. The comparison between the core samples and the portions below the rind samples showed some differences, primarily due to the low water content of the latter portion. Mainly, the S portions showed lower water content and a higher nitrogen percentage than the C portions. Concerning the two different sizes examined, namely, 3 and 5 kg, only small differences were observed, indicating that the size (weight) does not affect the quality of Provolone del Monaco cheese. These results, characterizing the nutritional profile of Provolone del Monaco PDO should be promoted throughout atlases, leaflets, brochures, handbooks at events at the national and local level, such as local restaurants, typical local products markets, street markets, and during gastronomic and local festivals with cultural and artistic events to promote the territory. Promotion and dissemination

of knowledge of Provolone del Monaco PDO, should be encouraged from nutritional, cultural, and touristic point of views, in order to reinforce gastronomic heritage and promote food tourism in rural areas for rural and regional economic development, as well as to support short food chain at zero kilometers. Indeed, the results should represent a valid tool for promotion of socioeconomic development, enhancement of territories, biodiversity preservation, and sustainability.

**Author Contributions:** R.R., A.S., N.M. conceived the work. N.M., R.R., A.S., A.D., M.L., A.A. wrote the manuscript. N.M., F.P., A.M., G.M., A.A. carried out the experimental study and analyzed the data. All authors made a substantial contribution to revise the work and approved it for publication.

**Funding:** The research received no external funding.

**Conflicts of Interest:** The authors declare no conflict of interest.

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
