# Peer review of "Influence of Ripening on Chemical Characteristics of a Traditional Italian Cheese: Provolone del Monaco"

_sustainability, doi:10.3390/su11092520_

Reviewer 1 Report

To the authors of the manuscript entitled: Influence of ripening on chemical characteristics of a Traditional Italian Cheese: Provolone del Monaco.

I have read and reviewed the manuscript. The authors did a great job at explaining the origin and characteristics of a traditional Italian cheese (Provolone del Monaco). The manuscript is, in general well organized and written. Sections 2 and 3 need to be switched (the results and discussion are currently presented before the materials and methods).

The authors mention that, as a result of the ripening process, "a large production of nitrogen"... was observed (Abstract, Line 28). Would it be more appropriate to say "ammonia release"?

The english grammar can be improved a little. For example, line 35: ... changes in life style led up to take attention towards... it would make more sense if "take attention" is changed to "put" or "pay" attention. Line 48, change "In this perspective" for "From this perspective". Line 56, would "milk origin" be more appropriate than "milk quality"?. Line 58, edit sentence: "This cheese is cylindrical in shape.... 

I also suggest the authors report the protein and fat contents on a dry basis. That would help them explain the results better. It definitely doesn't make sense to say that the "fat content increased with aging". The concentration changes because moisture was lost in the process. 

The authors do a good job at describing the volatile and other compounds generated during aging. But should give more emphasis to the production of butyric acid (a very important aroma compound in this type of cheese). 

Line 395, did the authors mean to say "proteolytic"? It reads "protolithic"

Author Response

The authors want to express their gratitude to the reviewer for his/her valuable comments and suggestions. The authors’ replies to the individual points raised are reported in Italic below.

Review Report Form

Open Review

English language and style

( ) Extensive editing of English language and style required 
(x) Moderate English changes required 
( ) English language and style are fine/minor spell check required 
( ) I don't feel qualified to judge about the English language and style 

Linguistic revision of whole manuscript was carried out.

Yes

Can be improved

Must be improved

Not applicable

Does the   introduction provide sufficient background and include all relevant   references?

(x)

( )

( )

( )

Is the research   design appropriate?

(x)

( )

( )

( )

Are the methods   adequately described?

(x)

( )

( )

( )

Are the results   clearly presented?

( )

(x)

( )

( )

Are the   conclusions supported by the results?

(x)

( )

( )

( )

Comments and Suggestions for Authors

To the authors of the manuscript entitled: Influence of ripening on chemical characteristics of a Traditional Italian Cheese: Provolone del Monaco.

I have read and reviewed the manuscript. The authors did a great job at explaining the origin and characteristics of a traditional Italian cheese (Provolone del Monaco). The manuscript is, in general well organized and written. Sections 2 and 3 need to be switched (the results and discussion are currently presented before the materials and methods).

As suggested, Material and Methods were presented before Results and Discussion.

The authors mention that, as a result of the ripening process, "a large production of nitrogen"... was observed (Abstract, Line 28). Would it be more appropriate to say "ammonia release"?

As suggested,  the text was modified.

The english grammar can be improved a little. For example, line 35: ... changes in life style led up to take attention towards... it would make more sense if "take attention" is changed to "put" or "pay" attention. Line 48, change "In this perspective" for "From this perspective". Line 56, would "milk origin" be more appropriate than "milk quality"?. Line 58, edit sentence: "This cheese is cylindrical in shape.... 

As suggested, “take attention” was replaced by “pay attention”, "In this perspective" by "From this perspective", "milk quality" by "milk origin". The sentence in line 58 was modified into “: "This cheese is cylindrical in shape.... â€ś.

I also suggest the authors report the protein and fat contents on a dry basis. That would help them explain the results better. It definitely doesn't make sense to say that the "fat content increased with aging". The concentration changes because moisture was lost in the process. 

Yes, we are agree with you. As suggested protein and fat contents were reported on a dry basis and related text was modified or deleted.

The authors do a good job at describing the volatile and other compounds generated during aging. But should give more emphasis to the production of butyric acid (a very important aroma compound in this type of cheese). 

As suggested, in the text more emphasis was given to production of butyric acid.

Line 395, did the authors mean to say "proteolytic"? It reads "protolithic"

Yes, "protolithic" was replaced by "proteolytic".

Submission Date

05 March 2019

Date of this review

22 Mar 2019 04:52:29

Reviewer 2 Report

The manuscript entitled „Influence of Ripening on Chemical Characteristics of a Traditional Italian Cheese: Provolone del Monaco” presents interesting issues, but it is not within the scope of the journal.

The article concerns chemical characterization of Provolone del Monaco cheese. Authors did not emphasize the association of the study with the aim of the journal. In the introduction authors stated that “promotion of local foods is well addressed towards a sustainable and environmentally friendly production systems” but the study is associated exclusively with the chemical composition.

In my opinion, this article, regardless of the quality, does not suit to this journal. Authors should submit this article to Food (MDPI), or anywhere else.

The section no 2 should be “Materials and Methods” instead of “Result”

Materials and Methods:

-        More information about materials are needed – e.g. what was the number of the samples? 

-        How many batches were analyzed? This is a crucial information
which affects the reliability of the results.

-        Line 381 – It should be “(α = 0.05)” or “(p ≤ 0.05)” instead of “(p < 0.05)”

Results and discussion:

-        Line 124 – It should be “(P ≤ 0.05)” instead of “(P < 0.05)”

-        Table 1 – The heading “Fat/Dry matter“ is unclear. Maybe “Fat to Dry matter ratio” will be better?

-        Taking into account that during ripening the Provolone del Monaco cheese loses water, the presentation of fat content in % could be misleading. It will be better to recalculate the data into g/ 100g of dry matter.

-        Tables 2 and 3 – the “different letters” in table should be in the superscript

-        Figure 2 is of poor resolution. It is unclear for what type of samples the PCA was presented. The point instead of coma should be applied for percentage (e.g., 74.73% instead of 74,73%).

Conclusions:

-        The conclusions are not related to the scope of the journal due to the fact the study is not directly related to the suitability.

Author Response

The authors want to express their gratitude to the reviewer for his/her valuable comments and suggestions. The authors’ replies to the individual points raised are reported in Italic below.

Review Report Form

Open Review

English language and style

( ) Extensive editing of English language and style required 
( ) Moderate English changes required 
( ) English language and style are fine/minor spell check required 
(x) I don't feel qualified to judge about the English language and style 

Yes

Can be improved

Must be improved

Not applicable

Does the   introduction provide sufficient background and include all relevant   references?

( )

(x)

( )

( )

Is the research   design appropriate?

( )

( )

(x)

( )

Are the methods   adequately described?

( )

( )

(x)

( )

Are the results   clearly presented?

( )

( )

(x)

( )

Are the   conclusions supported by the results?

( )

( )

(x)

( )

Comments and Suggestions for Authors

The manuscript entitled „Influence of Ripening on Chemical Characteristics of a Traditional Italian Cheese: Provolone del Monaco” presents interesting issues, but it is not within the scope of the journal. The article concerns chemical characterization of Provolone del Monaco cheese. Authors did not emphasize the association of the study with the aim of the journal. In the introduction authors stated that “promotion of local foods is well addressed towards a sustainable and environmentally friendly production systems” but the study is associated exclusively with the chemical composition. In my opinion, this article, regardless of the quality, does not suit to this journal. Authors should submit this article to Food (MDPI), or anywhere else

This  paper is   addressed to the peculiar Special Issue within Sustainability, MDPI “The Close Linkage between Nutrition and Environment through Biodiversity and Sustainability: Local Foods, Traditional Recipes and Sustainable Diets", where the valorization of local foods by identifying their nutritional quality and safety characteristics is one of main focus in line with our paper as well as the evaluation of the influence of factors on food
quality
.

As already underlined in the paper the nutritional characterization of local and traditional products is one of main actions for the valorization and promotion of traditional products. The envisaged promotion of local products contributes to environmental protection and it is a valid tool for promotion of socioeconomic development, enhancement of territories and biodiversity preservation.

The section no 2 should be “Materials and Methods” instead of “Result”

As suggested, Material and Methods were presented before Results and Discussion.

Materials and Methods:

-        More information about materials are needed – e.g. what was the number of the samples? 

As suggested, the number of samples was reported in the text.

-        How many batches were analyzed? This is a crucial information 
which affects the reliability of the results.

As suggested, the number of batches was reported in the text.

-        Line 381 – It should be “(α = 0.05)” or “(p ≤ 0.05)” instead of “(p < 0.05)”

The statistical analysis was performed for P<0.05.< span="">

Results and discussion:

-        Line 124 – It should be “(P ≤ 0.05)” instead of “(P < 0.05)”

The statistical analysis was performed for P<0.05< span="">

-        Table 1 – The heading “Fat/Dry matter“ is unclear. Maybe “Fat to Dry matter ratio” will be better?

We modified the table: have reported the results as  fat content (g / 100 g dw)

-        Taking into account that during ripening the Provolone del Monaco cheese loses water, the presentation of fat content in % could be misleading. It will be better to recalculate the data into g/ 100g of dry matter.

As suggested we have deleted the column showing  fat content in % and we have recalculated the data into g/ 100g of dry matter.

-        Tables 2 and 3 – the “different letters” in table should be in the superscript

The different letters in Tables 2 and 3 were reported as superscripts.

-        Figure 2 is of poor resolution. It is unclear for what type of samples the PCA was presented. The point instead of coma should be applied for percentage (e.g., 74.73% instead of 74,73%).

The resolution of Figure 2 was improved and to indicate the percentage  the comma was replaced by point.

Conclusions:

-        The conclusions are not related to the scope of the journal due to the fact the study is not directly related to the suitability.

  As above

Submission Date

05 March 2019

Date of this review

08 Mar 2019 13:35:50

Round  2

Reviewer 2 Report

Authors made effort to improve the manuscript, however I have some minor comments:

-Table  3 – the “different letters” in table should be in the superscript

-Figure 2 – typos in word “Aldehydes”. Abbreviations (t90, t180, etc.) should be explained. 

-The conclusions could be improved (should be more related to the conducted research)

At last, my final suggestion is to emphasize more the connection of the topic to sustainability.

Author Response

The authors want to express their gratitude to the reviewer for his/her valuable comments and suggestions. The authors’ replies to the individual points raised are reported in Italic below.

Inizio modulo

Open Review

English language and style

( ) Extensive editing of English language and style required 
( ) Moderate English changes required 
( ) English language and style are fine/minor spell check required 
(x) I don't feel qualified to judge about the English language and style 

Yes

Can be improved

Must be improved

Not applicable

Does the   introduction provide sufficient background and include all relevant   references?

( )

(x)

( )

( )

Is the   research design appropriate?

(x)

( )

( )

( )

Are the   methods adequately described?

(x)

( )

( )

( )

Are the   results clearly presented?

( )

(x)

( )

( )

Are the   conclusions supported by the results?

( )

( )

(x)

( )

Comments and Suggestions for Authors

Authors made effort to improve the manuscript, however I have some minor comments:

-Table  3 – the “different letters” in table should be in the superscript.

As suggested, the different letters in Table 3 were inserted as superscript.

-Figure 2 – typos in word “Aldehydes”. Abbreviations (t90, t180, etc.) should be explained. 

The term of Aldehydes was corrected. Abbreviations were explained in the note.

-The conclusions could be improved (should be more related to the conducted research)

At last, my final suggestion is to emphasize more the connection of the topic to sustainability.

As suggested, in  the conclusion additional lines  were added in order to mark more the linkage with promotion of territory, biodiversity and sustainability.

Submission Date

05 March 2019

Date of this review

17 Apr 2019 17:21:58

Fine modulo